# Ultrasound-Assisted Extraction and the Encapsulation of Bioactive Components for Food Applications

**DOI:** 10.3390/foods11192973

**Published:** 2022-09-23

**Authors:** Nitin Mehta, Jeyapriya. S, Pavan Kumar, Akhilesh Kumar Verma, Pramila Umaraw, Sunil Kumar Khatkar, Anju Boora Khatkar, Devendra Pathak, Ubedullah Kaka, Awis Qurni Sazili

**Affiliations:** 1Department of Livestock Products Technology, College of Veterinary Science, Guru Angad Dev Veterinary and Animal Sciences University, Ludhiana 141004, India; 2Institute of Tropical Agriculture and Food Security (ITAFoS), Universiti Putra Malaysia (UPM), Serdang 43400, Malaysia; 3Department of Livestock Products Technology, College of Veterinary and Animal Sciences, Sardar Vallabhbhai Patel University of Agriculture and Technology, Meerut 250110, India; 4Department of Dairy Technology, College of Dairy Science and Technology, Guru Angad Dev Veterinary and Animal Sciences University, Ludhiana 141004, India; 5Department of Dairy Chemistry, College of Dairy Science and Technology, Guru Angad Dev Veterinary and Animal Sciences University, Ludhiana 141004, India; 6Department of Veterinary Anatomy, College of Veterinary Science, Guru Angad Dev Veterinary and Animal Sciences University, Ludhiana 141004, India; 7Department of Companion Animal Medicine and Surgery, Faculty of Veterinary Medicine, Universiti Putra Malaysia (UPM), Serdang 43400, Malaysia; 8Department of Animal Science, Faculty of Agriculture, Universiti Putra Malaysia (UPM), Serdang 43400, Malaysia; 9Halal Products Research Institute, Putra Infoport, Universiti Putra Malaysia (UPM), Serdang 43400, Malaysia

**Keywords:** bioactive compounds, ultrasound, extraction, encapsulation, food processing

## Abstract

Various potential sources of bioactive components exist in nature which are fairly underutilized due to the lack of a scientific approach that can be sustainable as well as practically feasible. The recovery of bioactive compounds is a big challenge and its use in food industry to develop functional foods is a promising area of research. Various techniques are available for the extraction of these bioactives but due to their thermolabile nature, there is demand for nonthermal or green technologies which can lower the cost of operation and decrease operational time and energy consumption as compared to conventional methods. Ultrasound-assisted extraction (UAE) is gaining popularity due to its relative advantages over solvent extraction. Thereafter, ultrasonication as an encapsulating tool helps in protecting the core components against adverse food environmental conditions during processing and storage. The review mainly aims to discuss ultrasound technology, its applications, the fundamental principles of ultrasonic-assisted extraction and encapsulation, the parameters affecting them, and applications of ultrasound-assisted extraction and encapsulation in food systems. Additionally, future research areas are highlighted with an emphasis on the energy sustainability of the whole process.

## 1. Introduction

In recent years, the application of plant-derived bioactive components has shown an incremental trend in food applications. With the advent of urbanization and changing consumer habits, food being consumed is largely changing [1,2]. More insistence on natural components has led to new strategies being devised for extracting these bioactives from natural resources. Even the wastes that are generated from plant and animal industries are looked upon as potential avenues for the generation of bioactive components [3,4,5,6]. Due to their benefits to human health as well as their antimicrobial and antioxidant properties in food matrices, these bioactive compounds have recently attracted a lot of attention in the food industry [7,8,9,10,11,12]. Nevertheless, differing conditions and storage times may have an impact on their bioavailability, bioactivity, and stability. One of the key attributes is the vulnerability of these bioactive components to adverse food environmental conditions during processing and preservation [13]. The active components largely become incapacitated or destroyed, leading to their weak action in the food matrix. Many technologies have been devised to protect and conceal these compounds into a matrix which results in protection and controlled release in food systems, yielding better results. One of the most significant applications in the food industry is the encapsulation of bioactives for the desired results. In addition to improving the stability of bioactive substances with controlled release, encapsulation technology also ensures enhanced sensory acceptability due to the masking of strong flavors.

Nowadays, the product quality as well as nutritional value go hand in hand. Consumers are demanding foods which have superior physical, chemical, microbiological, and nutritional qualities. The concept of functional foods and nutraceuticals is dominating the present food market. The technological interventions for the development of these foods are challenging and urgent. Encapsulating bioactive components of nutritional value can boost their stability as they move through the gastrointestinal tract and safeguard their activity during processing and storage [14]. The component that needs to be entrapped is referred to as the core, fill, or internal phase, which can be essential oils, plant extracts, flavonoids, pigments, enzymes, nutrients, etc., and the wall materials are referred to as the shell, capsule, coating, or external phase [13,15]. This fundamental idea, which is based on the core and wall models of cells, is insufficient to describe the various technologies that have been created, as well as the outcomes of those technologies. Therefore, today’s definition of encapsulation is a method of stabilizing active substances through the structuring of systems that can ensure the preservation of their chemical, physical, and biological properties as well as their release or delivery under predetermined or desired conditions. Encapsulation aids in the distribution of the bioactive substances and allows for a regulated discharge to occur at different/extended times. By using this method, the substance is better protected, safe, and effective [16]. The basic concept of microencapsulation of bioactive compounds is presented in Figure 1.

Prior to encapsulation, the extraction of bioactive components is an important step. The extraction can be carried out through conventional and non-conventional methodologies. Conventional methods have been in use for many decades but they have major limitations such as a longer extraction period, higher solvent consumption, degradation of thermo-sensitive components, longer processing times, high temperature application, more energy intensive, etc., resulting in the search for alternative non-conventional methods for better results and efficiency [17]. In this context, the application of an ultrasound for the extraction of bioactive components is considered as one of the promising alternatives and is emerging as a green technology due to the lower cost of operation, shorter operation times, and lower energy consumption as compared to conventional methods.

## 2. Ultrasound Technology: An Overview

Rising demand for minimally processed, healthy, and safe food products is appearing center stage in coming years worldwide, which means that non-thermal technologies are now entering and gaining relevance in the food sector. These technologies are regarded as more energy-efficient, environmentally friendly, and relatively cleaner processes [18]. The need for the development of a number of non-thermal approaches to food processing has been prompted by the demand for convenience foods of the highest quality in terms of natural flavor and taste and which are free from additives and preservatives [19]. An example of a flexible non-thermal technology is ultrasound, which has a variety of uses in the processing of food. It is prominent amongst the list of “emerging technologies” such as ultra-high-pressure processing, pulsed electric fields, supercritical fluid extraction, microfluidization, and ultraviolet light treatment [20].

Ultrasound is the term used for a sound wave or all acoustic energy that is above the human hearing threshold, i.e., 20 kHz [21]. Human audibility range is usually between 20 Hz and 20 kHz [22] and primates and other closely similar animals with upper audibility limits greater than 20 kHz can, however, can hear ultrasound waves. Application of ultrasound is diverse and has broad dimensions in food science. Due to its role in environmental sustainability, ultrasonication is widely used in industries and is referred to as a “green unique technology”. Typically, ultrasonication treatments vary within frequencies of 20 to 100 kHz and power densities of 10 to 1000 W/cm^2^.The ultrasonic wave is propagated by the movement of particles in the medium and moves through large amounts of material depending on its mechanical and physical characteristics, such as texture and structure [23]. The vibrational energy of ultrasound is created by ultrasonic transducers and is relatively practically applicable for liquids, dispersions, solid, and gaseous media. The ultrasound-based instrumentations have several influencing factors such as temperature, noise, and full material contact with ultrasonic transducers, which have an impact on the output data.

On the basis of the frequency range, the ultrasound has been categorized into power ultrasound (20–100 kHz), high-frequency ultrasound (100 kHz–l MHz), and diagnostic ultrasound (1–10 MHz). High-frequency ultrasound and diagnostic ultrasound has varied applications including soft tissue surgery, diagnostic imaging, etc. [24]. On the other hand, power ultrasonography has been applied in food processing operations including extraction of bioactive principles and encapsulation thereof [25]. Even in food processing, both high frequency and low intensity ultrasound as well as low frequency and high intensity ultrasound have found their zones. High intensity and low frequency ultrasonic waves are disruptive in nature and significantly impact the mechanical and physical characteristics of food products. The frequency range of these waves lies between 20 and 100 kHz and intensities between 10 and 1000 W/cm^2^. Out of numerous applications of these waves, emulsification, microstructure alteration, textural modification and sonocrystallization, are the important ones [26].

## 3. Fundamentals and the Mechanism of Ultrasound-Assisted Extraction and Encapsulation

Ultrasound-assisted extraction primarily involves the interaction of ultrasonic power and solvents to extract targeted bioactive components from the plant biomass [27]. The application of ultrasound has improved the yield of bioactives as compared to conventional extraction methods [27]. The ultrasonic waves consisting of compression and rarefaction result in the dislodgement of molecules from their original position. At higher intensity sound waves, the pulling force over dominates attraction resulting in the creation of cavitation bubbles which grow through coalescence and collapse during the compression phase resulting in a hot spot formation. The acoustic cavitation is the main principle behind ultrasound-assisted extraction. On the other hand, ultrasound-assisted encapsulation, which is ascertained through emulsification, also depends upon the effectivity of the cavitation mechanism. The ultrasonic equipment promotes cavitation wherein bubbles collapse at the interface of both the immiscible liquids [28].

The material to be sonicated may be subjected to either direct or indirect ultrasonic treatments. The term “direct application” refers to the direct insertion of an ultrasonic probe into a liquid medium. The ultrasonic probe consists of a probe or horn connected to transducer. It is an important method for the extraction of active principles either from plants or processing byproducts. The probe diameter is an important factor contributing to the efficiency of the ultrasonic apparatus. Generally, the smaller diameter probes produce a greater cavitation effect but in a narrow field and on the contrary, the larger diameter probes distribute the energy to a wider area. A major disadvantage of this method is potential metal contamination from the probe’s detachable metal fragments. Ultrasonic baths, which are an example of “Indirect application” are more economical and easier to handle but they are not powerful equipment; hence, their potential applications are severely constrained [29]. Further the non-uniform energy distribution reduces its working efficiency as compared to probe system.

As far as the generation of ultrasound is concerned, the most fundamental tool used to create ultrasound is a transducer, which transforms electrical pulses into the necessary amount of intense sonic energy [30]. Irrespective of type of sonication, i.e., probe or bath, the ultrasonic system is essentially comprised of the generator, transducer, and application system. The key role of the generator is the transformation of electrical energy into desired alternating currents at an ultrasonic frequency which drives the transducer [31]. The transducer is the main device which converts the current into mechanical vibrations. Ultrasonic transducers are generally classified into piezoelectric, magnetostrictive, and fluid-driven transducers [32]. By applying pressure to a thin blade made of metal, the fluid-driven transducer generates vibrations at ultrasonic frequencies that can be employed in mixing and homogenization systems. To create the magnetostrictive transducer, a type of ferromagnetic material is used. When a magnetic field is applied, these changes cause desired mechanical vibrations. The system’s efficiency is only about 60% of transfer to acoustic energy [33]. By altering the size of piezo ceramic materials such as lead zirconate titanate, barium titanate, and lead metaniobate in response to electrical inputs, piezoelectric transducers generate acoustic energy. This is the most widely used and effective technology that leads to 80 to 95% percent transfer to acoustic energy [34].

The fundamental application of ultrasound in encapsulation technology is related to its role as an emulsification method because of the intense shear produced by acoustic cavitation as a sequalae of the application of ultrasound. The emulsification is rather a simple technology wherein the droplets are broken down into small structures leading to an increase in surface area and then the stabilization of those droplets is carried out by the action of emulsifiers. It ensures that there is no coalescence and creaming which may lead to a deterioration of emulsion quality [35]. Ultrasound-assisted encapsulation through emulsification usually encounters the problem of ‘overprocessing’ which may lead to an increase in droplet size due to recoalescence. It generally happens when the intense acoustic waves break the droplets into smaller ones, but the emulsifier is not able to quickly act leading to opposite results of droplet aggregation. Thus, the ultrasonication parameters need to be standardized along with emulsifier concentration to produce desirable results.

Various aspects of ultrasound-assisted encapsulation are presented in Figure 2.

## 4. Parameters Involved in Ultrasonic Extraction and Encapsulation

The power, frequency, and intensity (UI) are attributes of ultrasonic waves whereas ET, shape, and size are related to ultrasonic equipment [36]. Ultrasound power refers to the rate of sound energy emitted per unit of time as noted in Equation (1). It is affected by the solvent mass (m), solvent heat capacity (Cp), and variation with time (T) [37].
P = m × Cp × dT/dt(1)

Frequency is an important factor in modulating the physical and biochemical effects of bubble collapse and shortening the refraction state, thereby boosting cavitation during ultrasound-assisted extraction [38].

Ultrasound intensity (UI) refers to the power delivered by the emitting ultrasonic source per unit as described in Equation (2) [38]. Minimizing the UI is desirable to achieve the cavitation threshold after which the higher acoustic pressure causes agitation and consequently the loss of ultrasound wave propagation and cavitation [29].
UI = P/S(2)

### 4.1. Frequency

For ultrasonic-assisted extraction, low frequency high intensity ultrasound results in a strong shear and mechanical force which is desirable for extraction; however, high frequency sound waves produce a large number of reactive radicals [39]. Though a number of studies have been carried out visualizing the effect of varying frequency on the extraction yield as well as quality, but most of them concluded that operations should be carried out at a constant frequency. A constant low frequency is preferred due to the formation of fewer of cavitation bubbles with a comparatively greater diameter yielding a larger cavitation effect as compared to the higher ultrasonic frequency [40]. Large sized cavitation bubbles are a favorable event during ultrasonic extraction which depends on the duration of the compression and rarefaction cycle. If this cycle is too short, the bubbles with smaller diameters will be produced resulting in a poor extraction yield.

Ultrasonic frequency and the cycle duration are inversely related to each other, signifying that at a higher frequency, cavitation bubbles will have much less time to grow, hampering their implosion effect and increasing their number which ultimately results in more resistance to mass transfer [41,42]. The effect of varying the frequency, i.e., 19, 25, 40, and 300 kHz on the UAE of oil from soybean germ was analyzed with the conclusion that the best yield was obtained at 19 kHz [43]. Similarly, three different frequency ranges, viz., 40, 40, and 120 kHz were applied for the extraction of phenolics from grape pomace with the conclusion that out of all the frequencies, the best results were obtained at the lowest one. Hence, in general, most of the extraction is carried out at lower and constant frequencies of 20 kHz as has been completed in the extraction of pectin from pomegranate peel [44], mango peel [45], orange peel [46], and grapefruit peel [47].

In ultrasonic encapsulation processing, i.e., emulsification, cavitation commonly happens at frequencies of about 20 to 100 kHz. The core ability of ultrasound for emulsification largely depends on the frequency used. When the frequency is increased, more energy is used to develop the emulsion [28]. As a result, several investigations successfully encapsulated bioactive substances utilizing frequencies that were closer to the lower limit of the previously specified range. Using ultrasonic technology, the microencapsulation of turmeric oleoresin at a frequency of 20 kHz in gelatin–collagen matrices was carried out. The ultrasonication was performed in a double jacketed container with water circulation to keep the temperature at 40 °C [48]. According to the authors, encapsulating turmeric oleoresin with ultrasound technology at a frequency of 20 kHz proved successful. The greatest outcomes, as evident by the smallest particle size, the higher retention of curcumin, and a stronger color stability under light incidence, were shown when sonication and spray-drying were combined. This study concluded that high-intensity ultrasound results in the production of microcapsules encapsulating curcumin with a better ability to function as food supplement or coloring dye.

On the contrary, the majority of research studies on the emulsification of food-related products used the ultrasonication mentioned frequencies around 20 kHz [49,50]. Encapsulation of annatto seed oil at a frequency of 19 kHz produced droplets with a mean size of only 0.7 mm [49] and the encapsulation efficiency was close to 90%, resulting in relatively stable emulsions without any creaming in 24 h. The encapsulation of oregano leaves extract induced intense cavitation at the same frequency of 19 kHz that allowed for the retention of about 84 percent of the thymol [51]. Because of the vigorous nature of the bubble collapse, high shear forces are generated at these low frequencies, which leads to good outcomes. Effect of multifrequency ultrasound on zein-encapsulated resveratrol was studied using triple frequency simultaneous ultrasound (20/28/40 kHz) and it was reported that it led to the highest encapsulation efficiency and loading capacity as compared to the single frequency. Even the double/dual frequency was better than the single one as evident by the significant reduction in the particle size and polydispersity index and the increase in the absolute value of zeta potential [52]. Appropriate frequency condition is an important ultrasonic control parameter which decides the efficiency of extraction as well as encapsulation through the ultrasound approach. Thus, for the best results, appropriate frequency conditions must be adopted.

### 4.2. Ultrasonic Power and Intensity

During UAE, power delivered to the extraction liquid medium is assessed by amplitude percent and 100% amplitude indicates the rated power of equipment [53], so amplitude % and power have a linear relationship. Power applied for extraction depends upon raw material matrix and the component to be extracted and it varies in range from 20 to 700 W [54,55]. The extraction yield increases with the increase in power applied but decreases after a certain level. The increased power application results in an increase in the size of cavitation bubbles which later collapse violently. The increased impact on implosion leads to fragmentation and pore formation in plant tissue material leading to increased diffusivity and extraction yields [56,57]. Moreover, the increase in power results in increased mechanical vibration which eventually increases the contact area of the solvent and tissue material leading to an enhanced penetration of the solvent, leading to a higher extraction yield. The interaction between power, time, and solvent is the most important factor in UAE. The linear effect of power on extraction yield has been reported wherein an increase in power from 100 to 200 W resulted in an increased yield of phenolic components from waste spent coffee grounds but power beyond 250 W resulted in a decreased yield of phenolic compounds [58].

Sonication power is another important factor which determines the efficiency of the encapsulation process. Ultrasonic power is basically the ultrasound generator’s output and defines the ability to interact with the material and alters its physical and chemical properties primarily through the cavitation process. For emulsification and ultrasound-assisted encapsulation through nanoemulsions, the ultrasonic power along with time of application plays a vital role [59]. In order to achieve droplets with the minimum size in the shortest possible time, the optimization of ultrasonic power is a must. An increase in ultrasonic power results in a decrease in sonication time till the end effect is noticed but beyond a certain limit, coalescence can develop and hence the increase in power does not signify the efficiency of the process [60]. The power values for small scales (lab or pilot scales) typically vary from 100 to 1000 W. It is also essential to keep the total energy of the system constant which is essential to encapsulate the bioactives without any appreciable reduction in activity potential. It is a well proven fact that applied or generated power alone will not have much significance in discussions and it would be meaningful to consider power which is delivered from surface of tip of probe, known as ultrasonic intensity and total energy dissipated, known as energy density into the sample [61].

Ultrasonic intensity is expressed as W/cm^2^ that measures the power transmitted to the medium through the probe tip [62]. It refers to how intense the vibration produced by the power source will be and consequently the extent of cavitation. The amplitude of the ultrasonic source’s vibration determines the power of sound waves, and as the amplitude increases, there is an increase in vibration intensity and its sonochemical effects [29]. At higher power levels, the conversion of applied power to delivered power was less efficient which signifies that the increase in ultrasound intensity results in a reduction in the conversion of applied power into delivered power. This could be due to the increased production of acoustic bubbles near the tip of ultrasonic horn which might have isolated the remaining solution from the impact of ultrasound waves [63]. To allow encapsulation by the emulsification process, a threshold value of ultrasonic intensity should be reached. More than 10 W/cm^2^ of intensity results in effective cavitation and some studies reported better results for the encapsulation of target components at high-power intensity values [64]. Though power intensity provides a comparatively accurate assessment of power delivered and transmitted into the system, for the purpose of comparison with other techniques such as microfluidization, high pressure processing, etc., the measurement of energy density is rather preferred.

Lower energy densities are required for processing coarse emulsions with identical characteristics while using ultrasonication and energy densities can vary depending upon the system processed, i.e., depending upon apparent viscosities, and the volume treated, e.g., thymol-containing oregano extract was encapsulated through emulsification using ultrasonication and the researchers applied 7980 J/cm^3^ energy density for a median volume of 40 mL. A large energy density was used as the apparent viscosity of oregano extract was higher [51]. On the other hand, annatto seed oil was encapsulated by emulsification with ultrasonic waves using 512 J/cm^3^ of energy density [65]. This indicated that variations in energy densities were observed depending upon the encapsulating materials. As energy density increases, the whole encapsulation process becomes expensive and there will be a limit of the volume of processing sample; however, beyond a certain limit, the increase in energy density is technologically derogatory due to the formation of aggregates and clumps in coarse emulsion/solution to be sonicated leading to poor stability. Thus, a balance of energy density and power needs to be maintained while optimizing the parameters for encapsulation.

### 4.3. Processing Time and Temperature Control

Increase in the time of sonication increases the extraction yield initially but with further increases in the period of sonication, the yield decreases. Initial increase in time may result in an enhanced exposure of extracting solvent to plant biomass and increased fragmentation and sonoporation may lead to a release of solute in the extracting medium. However, beyond a certain limit, a higher sonication time leads to the structural destruction of the active component and results in a decreased extraction yield. Many studies have established the linear relationship of sonication time with extraction yield [66,67,68]. The effect of temperature on extraction yield is also similar to that of processing time and ultrasonic power. With an increase in temperature, extraction yield increases initially but decreases eventually. Temperature increase has a dual effect on the solute and solvent [39], wherein higher temperatures lead to a decrease in the viscosity of solvent and hence the increased penetration of the solvent is ensured in the tissue matrix. Moreover, an increase in temperature increases the solubility of the active component or solute extracted in the solvent.

The extraction yield of the total phenolic compound from grape seed increased with the increase in temperature with the maximum yield obtained at the highest temperature tested from 40 °C to 60 °C, with the maximum yield obtained at 52.8 °C [69]. A similar trend was observed for the total phenolic compound in black chokeberry waste wherein temperature (30–70 °C) was one of the independent variables and the extraction yield was maximum at the highest temperature tested, i.e., 70 °C (30 °C) [70], coconut (*Cocos nucifera*) shell powder (30 °C) [71], and the extraction of pectin from grape pomace in which extraction temperature as independent variable (35–75 °C) was taken and the highest pectin yield (∼32.3%) was achieved when the UAE process was carried out at 75 °C [72]. After the maximum extraction, a further increase in temperature leads to weak cavitation leading to a poor extraction yield. During the extraction of the phenolic compound from waste spent coffee grounds by UAE, the yield increased with the increase in temperature from 30 °C to 45 °C, and beyond 45 °C the yield decreased [73].

Another crucial parameter during ultrasound-assisted encapsulation through emulsification is the time of processing. Decision of time of processing and energy dissipated into the system is very important as short times result in the production of emulsions with a larger droplet size and higher processing times result in a large amount of energy investment into the system. Even longer times can cause poor stability as evident by coalescence and aggregation. Similar to the energy density range, the processing time also has a threshold value. Beyond that there is cavitational bubble cloud formation which diminishes the overall performance [63]. It has been reported in many studies that as the encapsulation by ultrasound largely depends on volume processed, type of equipment, its geometry and shape, the sole reliance on time of processing as a distinguishing factor for various methodologies will not be justified. So, the comparisons are made on energy density rather than processing time alone. Further, the behavior of droplet size with time, referred to as kinetics, is a better benchmark as processing time also modulates cavitation behavior.

The time of sonication also affects the temperature of the system as the collapse of the bubbles immediately leads to a severe increase in local temperature reaching values of thousand degrees [74]. It not only affects the efficiency of encapsulation process but can also initiate adverse reactions, e.g., while starch is utilized as a wall material, there can be gelatinization at higher temperatures, resulting in the poor performance of the whole system. The overheating of samples during ultrasonic emulsification should be appropriately controlled to prevent overheating, either by using a cooling mechanism or apparatus (slushed ice, etc.) or optimization of processing parameters. Another strategy is to use ‘pulse mode’ in sonication equipment [29]. The amplifier switches the power on and off for a certain period and this avoids an increase in temperature. Thus, all the parameters need to be simultaneously optimized and studied for effective encapsulation process by sonication.

### 4.4. Solvent Characteristics

Different polar (water) and non-polar (ethanol, acetone, etc.) solvents are used in UAE which affect the extraction yield. For the extraction of phenolic components from plant biomass, alcohols, and acetone, with varying levels of water, have been widely used. Out of all the solvents, ethanol is the preferred one for phenolic compound extraction due to its highest affinity with these components. Further, it also bears GRAS status and is readily available in the market [70]. Increase in ethanol concentration increases the extraction yield but thereafter a further increase has a negative effect on the yield. Ethanol leads to a decrease in the dielectric constant of the solvent which increases the solubility and diffusivity of phenolic components in the extraction solvent [6,75,76,77]. However, at higher concentrations, i.e., nearly 100%, it causes the dehydration of plant tissue and protein denaturation leading to diminished yields. This trend has been observed in the extraction of flavonoids from grapefruit solid waste in which the maximum extraction yield was obtained at ethanol concentration of 0.5 g/g and beyond that concentration, i.e., 0.8 g/g and 1 g/g, the total extraction yield of flavonoids decreased significantly [58] and cyanidin-3-O-glucoside extraction from jabuticaba peel in which extraction was favored by the increase up to 30% in ethanol concentration and beyond that, the increase in ethanol in the extraction solvent decreased the extraction of cyanidin-3-*O*-glucoside [78]. Contrary to the findings discussed here, increasing ethanol concentration at fixed time and temperature range, increased the extraction yield of phenolics from grape seed and was maximum at the highest ethanol concentration [69].

The research focus on the utilization of solvents for UAE has mainly centered around decreasing the effective concentration or totally eliminating it. Water has been utilized for the extraction of active components viz. phenolic, flavonoids and antioxidant from grape pomace but the aqueous UAE was able to extract only 12% to 38% of phenolic compounds from grape pomace as achieved by organic UAE [79]. Hence, a balance between concentration and other operating conditions is a big challenge. Lab and pilot scale ultrasound-assisted water extraction of polyphenols from apple pomace was conducted [80] and the optimized conditions obtained by response surface methodology for polyphenols water-extraction were 40 °C, 40 min and 0.764 W/cm^2^. As compared to conventional methods, UAE resulted in a 30% increase in the polyphenol extraction yield. Using water as an extraction solvent has yielded promising results in the extraction of gallic acid from from *Labisia pumila.* The result showed that extraction using water as a solvent yielded about 29% higher GA content (0.126 mg GA/g DW) than that of 10% ethanol (0.098 mg GA/g DW) [81]. Aqueous extracts from *Ascophyllum nodosum* brown edible seaweeds were obtained after continuous ultrasound-assisted extraction. Three different parameters (sonication time: 2 to 6 min; sonication amplitude: 80–100% and solvent/solid ratio: 20 to 40 g water/g) were taken into consideration. Extracts were characterized by polyphenols, carbohydrate, and uronic acid content and antioxidant capacities of extracts were measured employing different methods (DPPH, FRAP, and ABTS). The authors concluded that antioxidant capacity values (DPPH, ABTS, and FRAP) of extracts using water as solvent and sonication technology were comparable to those extracts obtained after very long extractions (>5 h) and employing organic solvents [82]. In a study, it was concluded that pure water as a solvent gave better extraction kinetics and higher extraction yields than pure ethanol or ethanol alcoholic mixtures with high ethanol percentage. This could be due to the fact that water can create a more polar medium that facilitates the extraction of phenolic compounds [83]. Another important dimension in UAE related to solvents is the liquid to solid ratio (LSR), which is the ratio between the extracting liquid and the quantity of sample taken for extraction. Similar to the trend followed for power, temperature, and solvent concentrations, the extraction yield of UAE increases with the increase in LSR up to a limit and decreases thereafter.

Lower LSR leads to a higher viscosity of the solution and the cavitation effect is relatively difficult due to more cohesion between particles of solvent whereas increasing LSR leads to a decrease in the viscosity of the extraction medium leading to a greater cavitation effect resulting in higher fragmentation and sonoporation, ultimately leading to better extraction yields. After a certain limit, this increase in LSR leads to enhanced cavitation resulting in the degradation of extracted bioactive component and solute, consequently decreasing the extraction yield [55,84,85]. During UAE of pomegranate peel, an increase in pectin yield was observed with an increase in LSR from 10:1 to 15:1 and the further increase in LSR to 20:1 decreased the pectin yield [44].

## 5. Ultrasound-Assisted Extraction

The key step during the recovery of bioactive components is extraction which can be carried out by many methods such as solvent extraction, supercritical extraction, microwave-assisted, extraction, etc. [39]. All these mechanisms have their relative disadvantages such as excess solvent requirement, low yield, capital intensive process, etc. [36]. Compared to these methods, ultrasound-assisted extraction (UAE) presents many advantages such as low cost of operation, lesser time requirement, energy efficient, retention of bioactivity in extracted components, etc. This technology is used for the extraction of flavonoids, polyphenols, carotenoids, etc., from the plant sources and offers promising results on an industrial scale.

UAE utilizes ultrasonic energy to extract target bioactive components along with the utilization of appropriate solvents [27]. Ultrasonic waves result in the formation of cavitaion bubbles through a series of compression and rarefaction cycles while passing through solids, liquids, or gaseous media. These bubbles may enlarge in size due to coalescence and collapse during the compression phase creating a hot spot, which may further accelerate the biochemical reactions in its vicinity [86,87]. Thus, the extraction from the source is mainly cased due to acoustic cavitation. The collapsing cavitation bubbles and sound waves result in fragmentation, pore formation and shearing in the cellular matrix of the plant along with other complex phenomenon which work singly or in unison, leading to the solubilization of the bioactive principle in the extracting liquid. Erosion caused due to ultrasound may also assist in the release of bioactive compounds. Sonoporation, i.e., pore formation due to cavitation also adds to the release of bioactive principles from cellular matrices [88,89]. The water absorption capacity of the material from which extraction has to be carried out is enhanced by the application of ultrasound. Thus, water acts as a solvent for the dispersion of the active component resulting in an increase in the yield. Application of ultrasound results in a higher yield of extraction with minimum loss of bioactivity.

Various dimensions of ultrasound-assisted extraction of bioactive compounds are presented in Figure 3.

Various advantages and disadvantages of ultrasound-assisted extraction of bioactive compounds are presented in Figure 4.

## 6. Applications

### 6.1. UAE of Proteins and Lipids

Ultrasonication has gained popularity in the extraction of proteins from different substrates and there are many reports of modification of physicochemical, structural, and functional properties of proteins on exposure to ultrasound. The protein extraction from defatted bitter melon seeds meal using pulsed ultrasound-assisted extraction (PUSAE) was carried out and the authors discovered improvements in water holding capacity, solubility, emulsion capacity, emulsion stability, and l gelation capacity in PUSAE protein samples as compared to conventional extracted bitter melon seed proteins. They also reported an increase in total protein yield [90]. By using sequential acid/alkaline isoelectric solubilization precipitation (ISP) extraction assisted by ultrasound [91], the optimized protein recovery from mackerel whole fish was studied and it was concluded that there was a possibility to increase the recovery yield from 49%. This new method also increased the extraction yield when compared to traditional ISP. Extraction of proteins from brewer’s spent grain (BSG) by using ultrasound-aided extraction (UAE) was studied, and it was concluded that more protein could be extracted as compared to conventional extraction procedures. In comparison to conventional extraction without ultrasonography, UAE working at 250 W for 20 min under a duty cycle of 60% may greatly increase the protein yield. Further, the UAE-treated BSG protein showed increased properties such as emulsifying, foaming, and fat absorption capacities which bear direct significance in fortification strategies. The authors concluded that UAE may be a potential protein-extracting technique which can have a greater applicability in the food sector [92]. As far as protein modification as induced by ultrasound is concerned, many researchers reported a change in plant cell structure and size, physical changes including protein denaturation and unfolding, change in protein size and microstructure, change in rheological properties, electrical conductivity, and protein color [93,94,95].

### 6.2. UAE of Oils and Extracts

Ultrasound-assisted extraction using ultrasonic bath with a frequency of 53 kHZ for the extraction of clove essential oil with high efficiency was performed. A central composite design was used to optimize the extraction parameters and it was reported that the most noteworthy factor which affected the UAE yields was the temperature employed during extraction. The active constituents were identified and were of great interest for a wide range of applications. The essential oils extracted by ultrasonication possessed potent antibacterial activity against both Gram-positive and Gram-negative microorganisms [96].

Ultrasound-assisted aqueous enzymatic extraction (UAAEE) was used for the extraction of *Cinnamomum camphora* seed oil and the investigation by researchers depicted that the nutritional properties of extracted C. *camphora* seed oil obtained using UAAEE method were far better than conventional means and they were comparatively healthier for human beings. The results revealed that UAAEE was a potential substitute to the traditional methods for extracting oils from oil bearing materials, including plants. The authors also submitted that *C. camphora* seed oil obtained has great potential in the food industry for the production of functional foods and nutritional supplements [97].

Ultrasound-assisted extraction (UAE) with ethanol as the solvent was performed for clove leaves (*Syzygium aromaticum*) by adopting a central composite design. Evaluation of the antioxidant capacity of extracts and yield of the same was performed. The results so obtained were compared with the conventional method of maceration. It was observed that compared to conventional extraction, the ultrasound method was significantly better in terms of extraction yield and antioxidant capacity [98].

The extraction of lipids from marine microalgae is comparatively easier due to different matrix from freshwater algae due to their soft cell membranes. Application of ultrasonic causes releases lipid by disruption and break down of cell membrane leading to release of lipids [99]. A high extraction yield of lipids (24.6%) was obtained from *Chaetoceros calcitrans* upon UAE treatment for 5 min at 2.45 MHz frequency and 1.4 kW power [100]. However, UAE does not always increase the yield of lipids from algae and it also has an effect on the structure of lipids; thus, it requires identifications [36]. Nogueira et al. [100] noted the increasing concentration of saturated fatty acids from 7.7% to 15.5% and polyunsaturated fatty acids concentration from 12.8% to 21.8% with a decrease in monounsaturated fatty acids from 79.5% to 62.7% upon UAE lipid extraction of *Chaetoceros calcitrans* algae.

Stevanato and da Silva [101] observed a higher yield (25.19%) of radish oil by using ultrasound (60 °C, 1 h and 12 mL/g) as compared to soxhlet extraction (15.2% yield). Similarly, a higher lipid yield under UAE (10 min, 600 W power, hexane: isopropanol 3:2) of flaxseed (9.98 mg/25 mg) was recorded by Metherel et al. [102] as compared to lipid yield under Soxhlet extraction (9.25 mg/25 mg). However, some studies did not report a significant increase in the yield of lipids upon UAE such as crambe [103], passion fruit [104] and *Moringa oleifera* [105]. However, UAE resulted in a reduced extraction time and a reduced energy consumption by destroying the cells [106].

### 6.3. UAE of Carotenoids and Polyphenols

Natural pigments, belonging to different groups such as carotenoids, anthocyanins, betalains, etc., are in great demand by the industry not only for color purposes but due to proven health benefits. To recover these pigments from plant biomass, it is essential to increase solvent–sample contact and reduction in mass transfer resistance [107]. Ultrasound-assisted extraction is a promising technology to recover these thermolabile bioactive components with enhanced yield and efficiency. UAE has been used to extract carotenoids from different fruits and vegetable byproducts such as peels, pomace, paste, leaves, etc. Ultrasound-assisted extraction of astaxanthin from dry cell mass (DCM) of *Phaffia rhodozyma* was carried out [108] using mixtures of DMSO and acetone with varying compositions and acetone alone as the solvent for extraction. Prediction of astaxanthin solubility using UNIFAC (Universal Quasi-Chemical Functional-Group Activity Coefficients) method and simulations of cavitation bubble dynamics were carried out. The authors concluded that ultrasonication helped to increase the extraction yield of astaxanthin. UAE of carotenoids from mango peel and paste and its influence on bio accessibility was carried out [109] and they observed that the yield of beta cryptoxanthin and beta carotene was improved by ultrasonication as compared to the control and a higher percentage of non-bioaccessible beta carotene was shown for the Control-peel (79.48%) and Control-paste (70.41%), and the percentage was lower in the UAE samples. They concluded that the use of UAE improved the yield as well as bio accessibility of carotenoids in mango peel and paste.

An efficient UAE method was used to extract natural pigments from waste red beet stalks. For the maximum extraction yield of pigments, the Box–Behnken response surface design (BBD) was applied. The authors concluded that extraction with ultrasound bears great potential for the extraction of pigments from waste red beet stalks as compared to conventional means [56]. Similarly, the extraction potential of ultrasound was well proved by a study carried out involving *Bougainvillea glabra* flowers using Box–Behnken response surface design. It was concluded from the results that ultrasonication has great potential to extract bioactive natural colorants from flowers [84].

Ultrasound-assisted extraction of total carotenoids for potential use as a natural colorant in bakery product from mandarin epicarp has been investigated. Response surface methodology was applied to maximize the extraction yield and it was reported that at a temperature of 60 °C for a time period of 60 min and with a solid to liquid ratio of 0.0004 g/mL, maximum carotenoid yield was obtained. Ultrasound was thought to be a promising technology for the extraction of natural colorants for prospective food applications [110].

Optimization of a pulsed ultrasound-assisted extraction (PUAE) technique for the retrieval of pomegranate peel extract with better yield and antioxidant activity was carried out and it was observed that the application of this technique could save the energy and cost of the production of total phenolics as compared to conventional extraction methods and continuous ultrasound-assisted extraction [111].

Ultrasound-assisted extraction of antioxidants from pomegranate peels and its further nanoencapsulation in emulsions was carried out [112]. Maltodextrin, whey protein isolate, and its combination were tried as wall materials. Nanoemulsions so formed were lyophilized and added in three different concentrations to soybean and mustard oil and changes in oxidative stability index during storage as compared to non-encapsulated extracts and synthetic antioxidants indicated that encapsulated extracts performed better in controlling oxidative changes due to the controlled release of bioactive components.

The recovery of total anthocyanins and phenolics from blueberry wine pomace using ultrasound-assisted extraction (UAE), was improved, leading to better recoveries of both the bioactive components. According to the study, ultrasonic extraction of desirable bioactive components from residues in the food industry is an effective, affordable, and environmentally friendly extraction technique [113].

Different types of solvents are being used to extract carotenoids during UAE and they are classified as green solvents and conventional organic solvents such as ethanol, methanol, petroleum ether, etc. The polarity of the solvent is of great importance while carrying out UAE. Ultrasound-assisted extraction of Cantaloupe peel with the use of the combination of polar and non-polar solvents for lipid soluble compounds extraction indicated that there was an enhancement of solubilization of non-polar carotenoids [114]. Similarly, the difference in the polarity of solvents at different concentrations was reported to have difference in extraction yield of carotenoids from carrot pomace [115].

Two different emerging techniques, i.e., pressurized liquid extraction (PLE) and ultrasound-assisted extraction (UAE) were compared for the extraction of phenolic compounds from Tahiti lime pomace at different temperatures (60–110 °C) and times (5–40 min) in PLE and ultrasound power (160–792 W), time (2–10 min), and solvent to feed mass ratio (20–40 kg solvent/kg dried pomace) in UAE. The effects on extraction yield, polyphenols concentration, and antioxidant capacity were assessed. Though, PLE was strongly impacted by temperature and extraction time, UAE showed a weak reliance on power, solvent to feed mass ratio, and time. Ultrasound-assisted extraction mechanisms were compared with maceration, and it was revealed that the total phenolic yield was influenced by mechanisms that only ultrasound can provide [116].

Polyphenol extraction from natural sources has been an area of immense interest for food processors due to the growing demand for natural antioxidants. The combination of ultrasonication and novel solvents, i.e., Natural Deep Eutectic Solvents (NADES) have been investigated for establishing the viability of the development of quick and long-lasting methods of extraction. It was reported that use of ultrasound and NADES resulted in the enhanced yield of antioxidants, a reduction in solvent consumption along with a decrease in the time of extraction [117].

Using a central composite design, the optimization of ultrasound-assisted extraction of antioxidants from orange peels was carried out. Ethanol (50%, *v/v*) was used as an extracting solvent for potent antioxidants and optimal conditions for extraction were finalized as 60 °C extraction temperature, 30 min time period and LSR of 15 mL/g. Thereafter, the extracted components were encapsulated in alginate-chitosan beads to regulate and monitor the release of antioxidants under gastrointestinal tract conditions. The authors reported that the average size of beads was 252 µm and the encapsulation efficiency was 89.2%. No interaction between the core material, i.e., extract and wall materials, i.e., alginate chitosan matrix was observed in the FTIR analysis. In vitro release studies depicted the release of antioxidants at a rapid pace initially and at a slower speed thereafter and followed simple Fickian diffusion. The authors proved that the encapsulation of orange peel extract substantiated the improvement in antioxidant delivery post gastrointestinal digestion and the encapsulated beads of antioxidants can have wider applications as natural bioactive agents in foods and pharmaceutical sectors [118].

### 6.4. Hybrid Combinations of Ultrasound

Using ultrasound as an extraction tool has many advantages over conventional methods such as solvent extraction in terms of mass transfer but some flaws are still obvious in this method too such as the degradation of primary or secondary metabolites [119]. Combination with other innovative techniques can be a promising solution which could enhance energy, momentum and mass transfer. Ultrasound can be combined with diverse techniques such as hydrotrops, ionic liquids, deep eutectic solvents, microwaves, supercritical fluids, pulsed electric fields, and many more, which could ultimately result in the enhancement of operational efficiency.

A comparison of extraction capabilities of ultrasonic and microwave-assisted extraction (UMAE) and ultrasonic-assisted extraction (UAE) of lycopene from tomato paste was completed [120], taking into account power, frequency, ratio of solvents to tomato paste and extraction time into consideration. The optimal conditions for UMAE were 98 W microwave power together with 40 KHz ultrasonic processing, the ratio of solvents to tomato paste was 10.6:1 (V/W) and the extracting time was 367 s. For UAE, the extracting temperature was 86.4 °C, the extracting time was 29.1 min and the ratio of the solvents to tomato paste was 8.0:1 (V/W). After processing, the percentage of lycopene yield in UMAE was significantly higher, i.e., 97.4% as compared to UAE, which was 89.4%. It implied that combination of ultrasound with microwave as an extraction tool was far more efficient than UAE alone.

It was shown that olive polyphenols rich in bioactive polyphenolic chemicals could be obtained using a practical sustainable method called cyclodextrin-enhanced pulsed ultrasonic-aided extraction (UAE). UAE use produced considerably better yields of bioactive polyphenols than conventional solvent extraction while also speeding up the extraction process and using less solvent [121].

Exploration of simultaneous Ultrasound–Microwave-assisted extraction (UMAE) system through a direct in situ configuration for solid–liquid extraction from sorghum husk was tried [122]. The authors reported that apart from an improved extraction yield, the extracts with better dyeing properties were obtained. Different bioactive components from almond seeds using pulsed electric field coupled with ultrasound treatment has been tried and it was demonstrated that both treatments resulted in higher value of total phenolics, total flavonoids, anthocyanin content and antioxidant activity as compared to other treatments [123].

Table 1 depicts the summary of processing protocols applied for extraction of bioactive compounds in the food industry.

## 7. Ultrasonic-Assisted Encapsulation

After the extraction of bioactive principles either from plant or its byproducts, it is imperative that these must be protected for the retention of their bioactivity in various food matrices. Ultrasound-assisted encapsulation is mainly carried out by emulsification which not only protects the active constituent but also expands the possibility of manufacturing functional products adding value to processing chain [49].

Emulsions can be defined as a mixture of two immiscible liquids wherein small spherical droplets of one liquid are dispersed into another. Out of them, one represents an oil phase (lipid phase) and other is an aqueous phase. The two phases are immiscible due to the difference in chemical nature and the contact of both liquids generates interfacial tension, resulting in thermodynamically unstable systems [150]. These are categorized into oil-in-water (O/W), water-in-oil (W/O) or multiple emulsions (W/O/W, O/W/O) on the basis of the composition of continuous and dispersed phases. Another classification of emulsions involves the measurement of droplet diameter of dispersed phase. The emulsion can be microemulsions and nanoemulsions (10–100 nm), miniemulsions (100–1000 nm) and macroemulsions (0.5–100 µm) [151]. Both oil-in-water (O/W) and water-in-oil (W/O) emulsions have high potential to encapsulate hydrophilic and hydrophobic bioactive components due to the presence of polar and non-polar moieties [152]. However, their kinetic stability is a decisive factor which is regulated by the addition of emulsifiers during emulsion preparation. Even the size of droplets plays an important role in deciding the stability and phase separation of developed emulsions. There are two different methods of producing emulsions, low energy methods represented by spontaneous emulsification and high energy methods which include high pressure homogenization and ultrasonication [153]. Application of both methods hold their own set of merits and demerits; however, the utilization of ultrasound as an encapsulating tool is gaining popularity at industrial level.

## 8. Application of Ultrasound-Assisted Encapsulation in Food Products

Ultrasound has been proven as a potent tool to encapsulate bioactive components through the emulsification or formation of emulsions. Thin emulsions with smaller droplet sizes improve the retention and encapsulation of bioactive compounds [154]. In this regard, low frequency high intensity ultrasound (16–100 kHz and 10–1000 W/cm^2^) assisted emulsification/encapsulation has excelled in the development of emulsions with qualities superior to those attained by conventional means [155,156]. The processing and encapsulation efficiency are increased with the use of ultrasound-assisted encapsulation which also increases shelf life and product stability. A number of bioactive components have been encapsulated using ultrasonication or with the assistance of ultrasound.

### 8.1. Encapsulation of Oils and Extracts

Application of ultrasound for the encapsulation of oils and extracts has been performed by many researchers [157,158,159,160]. Encapsulation of annatto seed oil using the emulsification technique assisted by ultrasound has been carried out along with assessment of stabilizing effects of inulin with different degrees of polymerization [156] and they found that ultrasonication leads to the production of viscous emulsion gels. Ultrasound-assisted emulsification was carried out using whey protein isolate and modified starch as adsorbents on an oil–water interface. The effect of ultrasonication power and time was visualized by droplet size and creaming stability. It was reported that the application of ultrasound resulted in the formation of fine and kinetically stable emulsions and out of the protein and modified starch, the former resulted in better encapsulation efficiency than latter [49].

By using an ultrasonic atomizer, fish oil encapsulation with chitosan was carried out [161]. Various variables for emulsion preparation viz. wall material concentration, i.e., chitosan, maltodextrin, and whey protein isolate (WPI), and the core component to be encapsulated, i.e., tuna oil, were optimized using an ultrasonic atomizer fitted with an ultrasonic probe of 1/32 in diameter that was operated at 40 kHz with 130 W high-intensity, and at 120 V. The researchers reported a higher concentration of active ingredients post encapsulation and concluded that the ultrasonic atomizer was the promising alternative method for tuna oil encapsulation.

Canthaxanthin was encapsulated in coconut oil-in-water emulsions using ultrasonic technology [162] who also used arabic and xanthan gums as emulsifiers. In order to produce emulsions with reduced average droplet diameter, the amounts of the three primary emulsion ingredients—gum arabic, gum xanthan, and coconut oil—were optimized in this work utilising response surface methodology. The authors claimed that by reducing the size of the droplets as well as by raising the density and viscosity of the oil droplets in the continuous phase, it was possible to improve the stability of the emulsion to phase separation. The canthaxanthin carotenoid’s encapsulation allows the construction of functional food products with this kind of natural component that can guarantee dietary benefits for human health and also aids in the creation of new sensory profiles for those new products.

Oil-in-water emulsions containing different essential oils were produced by using high intensity ultrasound technology (HIUS) and gum arabic as an emulsifier. The encapsulation efficiency and emulsion kinetic stability was assessed through an acoustic power and process time binomial approach, and it was found that emulsions had kinetic stability even after 72 h of storage except for orange oil nanoemulsions and high encapsulation efficiency. Thus, it was concluded that HIUS has the capability to produce stable emulsions for further use in various matrices [163]. Ultrasonic-assisted eucalyptus oil nanoemulsion was performed by using response surface methodology wherein the optimal sonication parameters were standardized. The assessment of particle size, zeta potential and anti-*E. coli* activity yielded ultrasonic emulsification as a novel technique for encapsulation of essential oil [164].

Soy protein isolate-maltodextrin-pectin complex stabilized pickering emulsions incorporated with Himalayan walnut oil was developed by ultrasound-assisted emulsification. Thereafter, fortification to develop functional mayonnaise was carried out and it was revealed that Mayonnaise enriched with omega-rich walnut oil encapsulated within the above-mentioned matrix resulted in positive attributes such as enhanced oxidative stability and higher storage modulus than loss modulus [165]. Ultrasonic-assisted encapsulation of oregano essential oil and extracts through development of oil-in-water (O/W) nanoemulsions as delivery systems has been studied. Thereafter its application in whey cheese was completed. It was observed that the combination of oregano essential oil and extracts resulted in nanoemulsions with effective antifungal and antioxidant activity and ultrasonication is considered a powerful method for encapsulation [166].

The stability of pandan (*Pandanus amaryllifolius*) extract encapsulated in microspheres with ultrasound assistance was studied [158]. All pandan-encapsulated microspheres demonstrated excellent stability after a month, with the exception of a minor size reduction for pandan in chitosan microspheres synthesized using 0.3 cm probe tip diameter. It is worth mentioning that both the size and distribution of the pandan-lysozyme and pandan-chitosan microspheres responded to the sonication parameters, particularly for their use in the food industry. The study involved indulgence of two parameters viz. ultrasonic probe tip and the core-to-shell ratio of 20 KHz ultrasonicator and the properties of encapsulates was studied. The authors concluded that modifying ultrasonic parameters could yield fine-tuned results for the encapsulation of the said extract with its stability during transportation. All of the pandan-encapsulated microspheres also demonstrated excellent stability after a month.

Microcapsules containing jackfruit extract were successfully made from a submicronic emulsion processed by ultrasound and then spray-dried and it was found that the obtained microcapsules were characterized by a good stability [167]. The bioactive compounds of the jackfruit extract associated with the microcapsules were protected since the efficiency of microcapsules for anti-proliferative and anti-mutagenic activity was significant. Using high-intensity ultrasound, encapsulation of sacha inchi (*Plukenetia volubilis*) oil in alginate and chitosan nanoparticles was studied [168] and the authors came to the conclusion that sacha inchi oil encapsulated in AL-CS nanoparticles demonstrated a higher loading efficiency and stability for short and long periods compared to other vegetable oils such as olive and soybean. Additionally, compared to nanocapsules containing olive oil, which also contain tocopherols, sacha inchi oil has significantly better antioxidant activity (95 percent radical reduction in 15 min).

### 8.2. Encapsulation of Flavor, Phenolics, Carotenoids and Pigments

The heat stability of cinnamon flavor has been increased by ultrasonic-assisted encapsulation for baking applications [169]. The interaction of encapsulated cinnamaldehyde with yeast for baking applications was found to be reduced. Microcapsules were formed by the cross linking of pectin and chitosan which was ensured by ultrasonication treatment. On analysis, it was found that as compared to pure cinnamaldehyde and un-sonicated formulation, there was a higher retention of cinnamaldehyde at high temperatures (>150 °C) for samples that were ultrasonicated. Lesser growth inhibition for *Saccharomyces cerevisiae* when subjected to encapsulated cinnamaldehyde was observed as compared to pure unencapsulated one. Encapsulation utilizing ultrasonication lead to higher heat stability to the encapsulated flavor resulting in greater retention of flavor at baking temperature range. Further, it also ensured lower concentration requirement of the flavor as compared to unencapsulated cinnamon flavor.

Ultrasonic atomizer was used for the encapsulation of natural astaxanthin in alginate-chitosan beads [170] and its application in yogurt was completed to improve the overall sensory profile. In order to create the emulsions, astaxanthin was mixed with aqueous phase alginate solution (0.8, 1.0, and 1.2 g/100 mL) in a mass ratio of 3:1 using a homogenizer at a rotational speed of 10,000 rpm for 15 min. Following that, the emulsion sample was pumped using a peristaltic pump into ultrasonic atomizer. The authors reported that encapsulation considerably increased the stability of natural astaxanthin and on the incorporation of encapsulated bioactive in yogurt, the customers rated it with 86.21 percent acceptability and 64.53 percent purchase intent.

It was reported that in baking applications, strong amide bonds created by ultrasonication- assisted cross-linking can improve stability of microcapsules encapsulating cinnamon flavor under high temperatures [169]. They further found a reduction in the interaction of encapsulated cinnamaldehyde with yeast and found that chitosan and pectin polymers were discovered to be suitable for creating microcapsules capable of absorbing the hydrophobic cinnamon taste in an aqueous continuous phase on formulation for encapsulating cinnamaldehyde using ultrasonication. They indicated that encapsulation following ultrasonication imparted greater heat stability to the encapsulated flavor.

### 8.3. Encapsulation of Nutrients

Development of functional foods by incorporation of ultrasound-assisted co-encapsulated probiotics and gamma amino butyric acid (GABA) was performed [171] by creation of double emulsion microcapsules. To stabilize the inner aqueous droplets of the double emulsion microcapsules, surfactants were used and the co encapsulation of both lactic acid bacteria and GABA was carried out. By adjusting the ultrasonication parameters to manage the size of the inner and outer emulsions, it was possible to co-encapsulate both LAB and GABA together. The authors concluded ultrasonication to be a potential tool for production of double emulsions encapsulating probiotics and further its targeted delivery to improve the functionality of food products.

Ultrasonic microencapsulation of oil soluble vitamins (A, D and E) was carried out using raw egg white proteins as shell materials [172]. The microcapsules so formed were protected from photodegradation by coating with green tea catechin/iron complex. The encapsulated vitamins were resistant to degradation upon cooking, storage and heating and were bioavailable by more than 2-fold as compared to free vitamins. On incorporation in food matrices, the encapsulated vitamins maintained their microstructures and provided a potential candidature in improvement of nutritional values of staple food products to combat maternal and child malnutrition.

Ultrasonic-assisted co-encapsulation and delivery of micronutrients using raw and native proteins as shell materials without any chemical denaturants for food fortification was studied [172,173]. Three different proteinaceous biopolymers, i.e., egg white protein, soy protein isolate, and corn protein isolate were shown to be able to create microcapsules by the ultrasound approach to encapsulate both oil- and water-soluble micronutrients. The authors concluded that proteins with high thiol content and weak intramolecular interactions provided better shell-forming ingredients for creating sturdy microcapsules for the encapsulation and distribution of micronutrients. Microcapsules with thick shells performed better in preventing the deterioration of micronutrients. Such protein-based microcapsules permitted the regulated release of micronutrients throughout an in-vitro digesting process, perhaps enhancing the absorption of nutrients in vivo. The created microcapsules are suitable for use in practical food applications, such as preventing malnutrition, due to the adaptability of the shell materials and core composition.

The effect of ultrasound-assisted zein-gum Arabic (GA) complex coacervates on the encapsulation of resveratrol was studied [174]. As compared with zein, zein-GA complex coacervation largely improved the encapsulation efficiency from 26.19% to 55.85% and after sonication it was relatively better, i.e., 74.2%. The loading capacities also increased three times, i.e., from 2.79% to 6.01%. The addition of GA led to an increase in particle size and polydispersity index which was reduced on application of the ultrasound. It was reported by authors through scanning electron and atomic force microscopy that ultrasonic treatment resulted in the formation of compact, homogenous composite particles due to the strengthening of hydrogen bonding and electrostatic interactions. This proved the role of ultrasound in enhancement of zein–GA complex coacervation to load resveratrol and depicted its effect in modification of the complex as delivery vehicle in functional food industry.

Ultrasound-assisted synthesis of encapsulates using natural and biodegradable glycogen nanoparticles derived from multiple sources such as bovine and rabbit liver, sweet corn, oysters, etc., for the encapsulation of soyabean oil and vitamin D has been studied. Various parameters for synthesis such as ultrasonic power, time, and concentration of glycogen and lipid phase were optimized. At a sonication time of 45 s and power of 160 W, native glycogen yielded encapsulates of diameter between 0.3 µm and 8 µm and these provide potential delivery vehicles for nutrients in various food processing applications [175].

Ultrasonic processing has also been frequently used for the modification of proteins for further application in encapsulation. Modification of pea protein isolates through ultrasonic processing in phosphate-buffered saline (PBS, pH 7.4) and Tris/HCl (pH 8) buffer systems for its prospective application in the encapsulation of functional liquids was studied [176]. Tetradecane-filled microspheres were synthesized using ultrasonically modified PPI as a shell material under high intensity 20 kHz ultrasound irradiation. Tetradecane was used as a model liquid, which could be replaced by functional liquids such as fish oil, etc. Solubility of water insoluble globulin fraction in PPI was significantly improved by ultrasonication as confirmed by sodium dodecyl sulfate-polyacrylamide gel electrophoresis (SDS-PAGE) analysis. Ultrasonic parameters were controlled to yield microspheres of PPi with desired size, distribution, shell thickness and encapsulation yield. During storage, the developed microspheres had stable structure and maintained size with smooth morphology even after one month at 4 °C. The processing protocols of some commonly used encapsulating materials with ultrasound is presented in Table 2.

## 9. Gaps and Future Scope of Research

Ultrasonic-assisted extraction and encapsulation has wide applications in food science as it offers huge advantages in terms of reduction in time, temperature, and energy for the process. However, the bigger challenges are process optimization with an outlook of keeping the bioactivity of extracted and encapsulated components intact for further food applications. In this context, total chemical solvent elimination during extraction without any implication on extraction yield is the area of research. Even utilization of various ancillary systems may help in increasing the efficiency and effectivity of extraction. Design of appropriate instrumentation is another dimension which can be worked upon to increase the effectivity of extraction. Mathematical modelling to understand complexities of cavitation process may help in optimizing the extraction parameters in future. Combination of ultrasound with ionic liquids and deep eutectic solvents may assist the food industry to establish a synergistic approach. Similarly, for encapsulation, controlling the energy input and making the whole process energy friendly is a big challenge. Behavior of core material encapsulated in various temperature and humidity conditions is must. Development of extraction and encapsulation processes as line operations in a food industry is future area of research but the main consideration is energy input and environmental protection. Thus, energy audit of both processes as compared to conventional techniques will give a fairer idea for the sustainability of these processes.

## 10. Conclusions

Generating the best out of waste is one of the most important components of sustainable development. The use of ultrasound-assisted extraction, which is primarily a novel nonthermal technology, is a boon for the food industry. Enhanced focus on devising energy intensive methods, judicious use of raw components, and recycling wastes to prevent environmental degradation has catalyzed efforts for development of this technology. The demand for natural bioactives is increasing day by day and hence extraction followed by encapsulation will help in solving the problems of the food processing industry at large. Proper optimization processes, devising instruments which are both effective and efficient, and continuous improvement in tech with focused research will open new vistas and will help solve the problems of the food industry with promising solutions.

## Figures and Tables

**Figure 1 foods-11-02973-f001:**
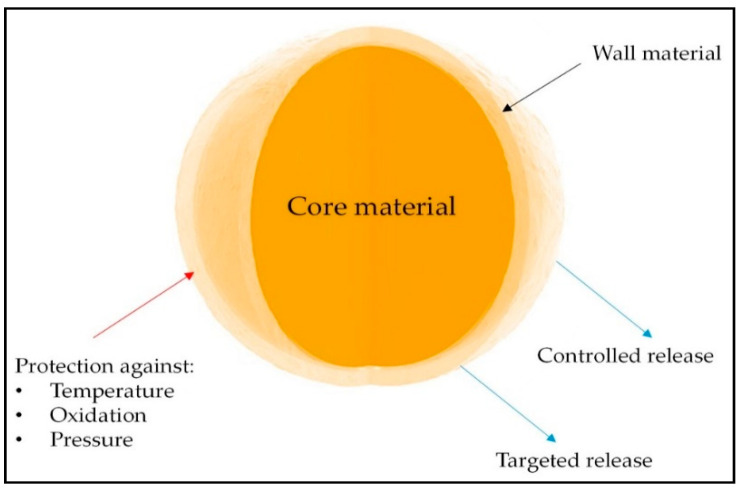
Basic concept of microencapsulation of bioactive compounds (adapted from [13] under CC BY license).

**Figure 2 foods-11-02973-f002:**
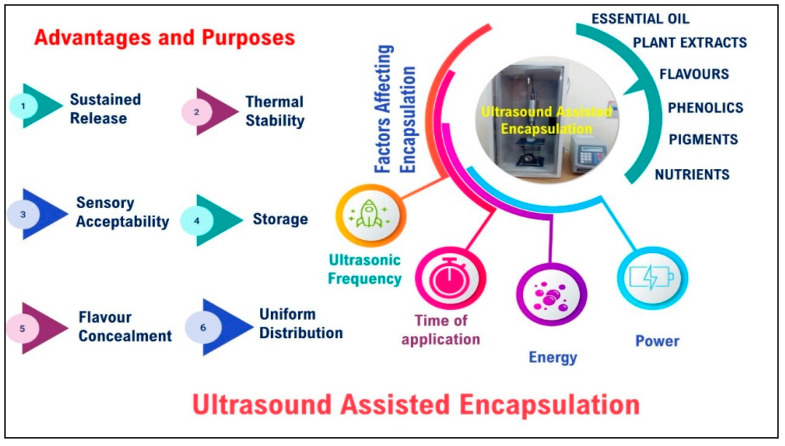
Various aspects of ultrasound-assisted encapsulation.

**Figure 3 foods-11-02973-f003:**
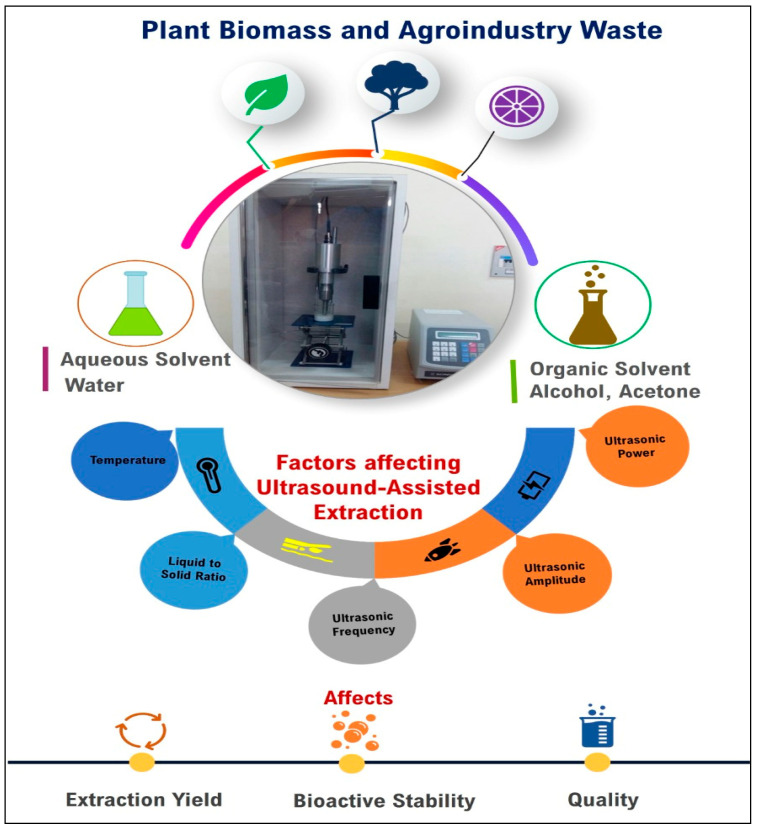
Various dimensions of ultrasound-assisted extraction of bioactive compounds.

**Figure 4 foods-11-02973-f004:**
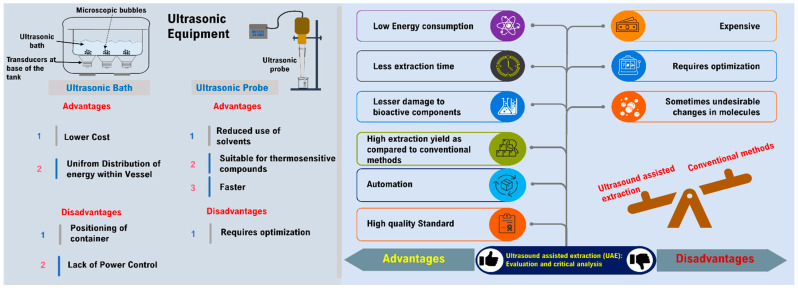
Various advantages and disadvantages of ultrasound-assisted extraction of bioactive compounds.

**Table 1 foods-11-02973-t001:** Various processing protocols applied for extracting food materials with ultrasound.

Extraction Method	Source	Bioactive Compound	Ultrasonic Device	Ultrasonic Parameters	Extraction Parameters	Yield/Measurements	References
Extraction of Phenolic Compounds							
Ultrasound-assisted extraction	Aronia grapes	Polyphenols		F: 20 kHz; T: 30 °C	t: 10 min	Color: 13.98 AU Total anthocyanin content: 140.15 mg/L Cyaniding-3-0-galactoside: 340.12 mg/L	[124]
Ultrasound-assisted Extraction	*Opuntiaficus indica*	Phenolic compounds	-	F: 50 kHz; P: 300 W	Maceration extraction: 5000× *g*; t: 20 min; T: 4 °C Soxhlet extraction: T: 85 °C	13.8 μmol Trolox Equivalent (TE)/g DW—DPPH, 4078 μmol TE/g DW—Ferric Reducing Antioxidant Power (FRAP), 3091 μmol TE/g DW—total antioxidant activity (TAA) 80.8 μmol TE/g DW for Trolox Equivalent Antioxidant Capacity (TEAC)	[125]
Ultrasound-assisted extraction	Cereal brans	Phenolics	Ultrasonic probe	F: 20 kHz; t: 10 min	Peleg’s model; centrifuged at 5000× *g*; t: 10 min	Total phenolic count: 1949 to 2152 mg/kg	[126]
Microwave- and ultrasound-assisted extraction	Black jamun pulp	Phenolic compounds Anthocyanin antioxidants	-	F:40 kHz; P: 100 to 220 W; t: 40 to 150 min	Microwave-assisted extraction Solid:solvent: 1:48 Centrifuged at 5000× *g*; t: 5 min	Phenolic components: 5.704 × 10^—12^ m^2^s^−1^; Anthocyanin: 2.485 × 10^−12^ m^2^s^−1^ Antioxidant activity: 2.061 × 10^−12^ m^2^s^−1^	[127]
**Extraction of Polyphenolic Antioxidants**							
Ultrasound-assisted natural deep eutectic method	Mango peel waste	Antioxidant	Ultrasonic probe	P: 0–200 W; F: 22 kHz; t: 30 min	Solvent: 80% ethanol; t: 60 min; T: 50 °C; duty cycle: 50%; acoustic density: 2 W/cm^3^; LSR: 30:1	DPPH: 35.3 µg/L TPC: 69.85 mg GAE/g of MP TFC: 16.5 mg QE/g of MP Particle size: 0.3 mm	[117]
Ultrasound-assisted extraction	Allium senesces L.	Polyphenol (Antioxidant efficacy in Harbin dry sausage)	-	P: 0, 150, 300, 450, 600 W; t: 5, 10, 15, 30, 45 min; T: 20 °C	Centrifuged at 5300× *g*; t: 10 min; T: 4 °C; Vacuum rotator evaporator	Moisture: 4 to 8 g/kg Carbonyl content: 6 to 8 g/kg Lipid oxidation: 6 to 8 g/kg	[128]
Ultrasound-assisted extraction	Turkey berry fruits	Phenolic antioxidant	Ultrasonic bath	F: 40 kHz; t: 3–5 min; T: 40–80 °C	Box–Behnken model Conventional solvent extraction, T: 4 °C, ethanol solvent: 40% to 80% *v/v*	TPC: 192.3 mg GAE/g, DPPH: 47.4 μg AA/g; ABTS: 116.2 μmol TE/g 56.7% ethanol concentration; 80 °C, extraction temperature; 17.3 min extraction time; 49.7 mL/g solvent-solid ratio	[129]
Ultrasound-assisted extraction (UAE)	Apple pomace (Freeze dried, fresh.macerated, one time and two time UAE)	Polyphenols (Beef burger fortification)	-	-	-	Yield: 7.12% to 13.61% Highest values of ABTS: 27.22 mg TE/g DW; DPPH: 10.50 mg TE/g DW; FRAP: 1.27 mg TE/g DW were found for freeze-dried extract; Total monomeric anthocyanin: 5.32 mg/L; Protein content: 2.70%; Fibre content: 40.19%; Esterification degree: 65.54%; beef burger fortified with apple pomace had lower water activity	[130]
**Extraction of Flavanoids**							
Ultrasound-assisted aqueous two-phase extraction	Jujube peel	flavanoids	Ultrasonic bath	P: 200 W; t: 20, 30, 40, 50, 60 min Solid:liquid ratio: 1:25, 1:45 g/mL	Centrifuged at 1616× *g* t: 15 min	Yield: 4.87 to 7.14 mg/g Main flavonoid: Rutin: 0.28 mg/g; particle size: 116.87 nm	[131]
Ultrasound-assisted extraction	Peanut shell	Flavanoids	Ultrasonic bath	P: 120 W; F: 45 kHz; T: 55 °C 70% ethanol: solvent	Extraction kinetics by peleg’s model and phenomenological model Soxhlet extraction: 98 °C; t: 320 min Heat reflux extraction: T: 85 °C; t: 320 min	Particle size: 0.285 mm Extraction yield: 9.263 mg/g Solvent to solid ratio: 40 mL/g	[132]
**Extraction of bioactive compounds**							
Ultrasound-assisted extraction	Red beet extract	Β cyclodextrin	-	F: 28 kHz; P: 80 W; without external heating	Centrifuged at 7000 rpm; t: 10 min	Betanin content: 2.243 mg Total phenolic content: 20.03 GAE/g DW DPPH activity: 59.87%	[133]
Ultrasound-assisted extraction	*Psidium guajava*	Guava leaves	Ultrasonic bath	t: 20 min F: 30 kHz	Hot water extraction 7000 rpm 5–20 min extraction time	Total polyphenolic content: 50.1–80.3% Antioxidant activity: 5.7–25.7 mg/100 g Vitamin C content: 44.7–289.77 mg QE/g Total flavonoids content: 0.02–0.2 g/mL	[134]
Ultrasound-assisted extraction	*Pelvetia canaliculata*	Sunflower oil	-	F: 35 kHz; t: 20 min	Centrifuged at 3000 rpm; t: 10 min	Protein content: less than 2% Carbohydrate content: 65% Chlorophyll pigment: 602 mg/kg Carotenoid pigment: 236 mg/kg DPPH activity: 86% FRAP activity: 3592% Flavanoids: quercetin: 7849 mg/kg; Catechin: 2966 mg/kg	[135]
Sequential microwave	Soymilk	Enzyme inactivity	Ultrasonic bath	Microwave ultrasound extraction P: 90 W; soybean:water = 1:5 t: 1, 2, 3, 4, 5 min	F: 28 kHz; t: 30, 60, 90 min	Protein content: 32.1% Viscosity: 3.31 mPa s Total soluble solids: 8.09 °Brix; pH: 6.5; Nitrogen content: 17.5% Fat content: 19% (db)	[136]
**Extraction of protein**							
Pulsed ultrasound-assisted extraction	Bittermelon	Protein	Ultrasonic processor	P: 450 W; t: 2.5, 5, 7.5, 20 min; T: 14 °C	Peleg’s model of extraction	Yield: 31.05% Protein content: 24% to 91% Water holding capacity: 2.31 g/g Foaming capacity: 27% to 68%	[90]
Ultrasound-assisted extraction	oleosin	Salt extraction	-	F: 20 kHz; t: 12 min; P: 0, 100, 400, 600 W	Centrifuged at 8000× *g*; t: 15 min	Oleosin yield: 17.6%; Particle size: 52 nm; DPPH activity: 242.08% ABTS activity: 240.71%	[137]
Ultrasound-assisted basic electrolyzed water extraction	Antarctic Krill (*Euphansiasuperba*)	Proteins	Ultrasonic probe	P: 350 W; F: 20 kHz; t: 0, 10, 20, 30 min	Centrifuged at 3000× *g*; t: 10 min	Particle size: 5–7 nm; yield: 9.4%; Purity: 78.37%; extraction yield: 68 to 77% Water absorption capacity: 4.96 g/g protein	[92]
Energy-efficient ultrasound extraction	Microalga C. vulgari	Protein	-	P: 1000 W; F: 20 kHz	T: 60 min; centrifuged at 2650× *g*	Protein yield: 48.1%	[138]
Ultrasound-assisted extraction	Sharpness stringray (*Dasyatiszugei*)	Pepsin and collagen	-	P: 750 W; t: 30 min	Collagen extraction Ultrasound-assisted extraction of acid soluble collagen (UASC) and ultrasound-assisted pepsin soluble extraction (UPSC): centrifugation at 6500× *g*; t: 15 min; T: 4 °C	Yield: UPSC: 61.50%. UASC: 48.37%, Moisture content: 7.77% to 8.09%, Protein content: UPSC: 89.89%; UASC: 89.81%, Fat content: UPSC: 3.96%; UASC: 4.16%	[139]
**Extraction of pectins**							
Ultrasound-assisted extraction	Sunflower by product	Pectin	-	t: 30 min; T: 33 °C; P: 400 W	Centrifugation at 10,000× *g*; t: 20 min Box–Behnken design	Yield: 11.15%; Water holding capacity: 5.37 g/g; Moisture: 9.06% Protein: 1.25% Ash: 1.43%	[140]
**Extraction of phytocompounds (Anthocyanin)**							
Microwave- and ultrasound-assisted extraction	Black jamun pulp	Phenolic compounds Anthocyanin antioxidants	-	F: 40 kHz; P: 100 to 220 W; t: 40 to 150 min	Microwave-assisted extraction Solid:solvent = 1:48 Centrifuged at 5000× *g*; t: 5 min	Phenolic components: 5.704 × 10^−12^ m^2^s^−1^; Anthocyanin: 2.485 × 10^−12^ m^2^s^−1^ Antioxidant activity: 2.061 × 10^−12^ m^2^s^−1^	[127]
Ultrasound-assisted extraction	Rosemary extract used as additive in Lingonberg pomace	Anthocyanin	Ultrasonic bath	F: 50 kHz; P: 80 to 200 W	RSM (Box–Behnken design)	Yield of anthocyanin 4.12 mg/g Cyanidin-3-galactoside yield of 3.36 mg/g, Cyanidin-3-glucoside yield of 0.15 mg/g Cyanidin-3-arabinoside yield of 0.61 mg/g	[141]
Ultrasound-assisted extraction	Black currant (*Ribesnigrum* L.)	Anthocyanin	-	F: 20 kHz t: 20 min	Microwave-assisted extraction; Box–Behnken design; Centrifugation at 50 to 200× *g*; T: 40 to 60 °C	Delphinidin 3–*O*–glucoside: 2.24 ± 0.14/g Cyanidin 3–*O*–glucoside: 9.07 ± 0.68/g Petunidin 3–*O*–rutinoside: 0.33 ± 0.01/g Peonidin 3–*O*–rutinoside: 0.28 ± 0.01/g Yield of anthocyanin from UAE method is higher than EAE method	[142]
**Extraction of oil**							
Ultrasound-assisted aqueous enzymatic extraction	Cinnamomum camphor seeds	Oil recovery	Ultrasonic bath	t: 30 min 6000 rpm	Plackett–Burman design and Box–Behnken design	Yield: 80.12% Acid value: 2.4 mg KoH/g Iodine value 2.86 to 2.9 g I_2_/100 g oil	[97]
Ultrasound-assisted extraction	*Moringa* *peregrina* oil	-	-	P: 348 W; T: 30 °C	Soxhlet extraction T: 50 °C; t: 4 h; solvent: hexane RSM (central composite design)	DPPH: 22.08% Peroxide value: 4 meq Iodine Value: 71.69 g Total phenolic count: 63.13 mg/g Yield: 53.10% Liquid:solid ratio: 5:1 to 20:1 (*w/v*)	[143]
**Extraction of carotenoids**							
Ultrasound-assisted extraction	Mandarin epicarp	Total carotenoids	Ultrasonic cleaner	F: 42 kHz; P: 240 W; t: 3.5 min	-	Solid liquid ratio: 0.004 g/mL Total carotenoids: 140.70 mg β-carotene/100 g	[110]
Ultrasound assisted extraction	*Lyciumbarbarum* L. fruit	carotenoids	Ultrasonic cell grinder	F: 20–25 kHz; P: 600 W	-	Total carotenoids: 132.75 mg β-carotene/100 g	[144]
**Extraction of saponins**							
Ultrasound-assisted extraction	Alfalfa (*Medicago sativa*)	Saponin	Ultrasonic cleaner bath	P: 50–150 W T: 50–80 °C t: 1–3 h	Ethanol: 60–90%	Yield of total saponins 18.6%	[145]
**Extraction of trace elements**							
Ultrasound-assisted extraction	Artichoke leaves and soybean seeds	trace metals (Cd, Cu, Ni, Pb, and Zn)	Ultrasonic bath and probe	Probe-based UAE: P: 750 W; F: 20 kHz; t: 10 min Bath-based UAE: F: 47 kHz; t: 25 min	Centrifuged at 28,000× *g*; t: 3 min (both probe and bath-based UAE)	Particle size: 200 µm	[146]
**Extraction of macrolides**							
Reverse ultrasound-assisted microextraction	Chicken fat sample	-	Ultrasonic probe	P: 91 W; t: 7.5 min; P: 130 W; F: 20 kHz	P: 91 W; t: 7.5 min; T: 75 °C; solvent:methanol, methanol/water, acetonitrile/water	Capillary electrophoresis analysis: 50 mmol L^−1^ LOQ: 17.4 to 55.0 µg kg^−1^ Timicosin Tylosin-22.1 to 47.0 µg kg^−1^ RSD%: 12.4%	[147]
**Extraction of food ingredients**							
Ultrasound prior to tertiary amine extraction	Milk phospholipid	Beta serum (dairy by products)	Ultrasonic horn	F: 20 kHz; t: 4 min	Folch extraction centrifugation at: 4200× *g*; t: 5 min; chloroform:methanol = 95:5 (*v/v*) chloroform:methanol:water = 5:3:2	Particle size: 579.7 nm to 163.1 nm Yield: 69%; phosphatidylinositol (32%); phosphatidylethanolamine (30%), and sphingomyelin (37%)	[148]
Ultrasound-assisted enzymatic extraction (UAEE)	Kiwi fruit	Starch	-	P: 300 W; T: 52 °C; solid:liquid ratio = 1:6.68	Box–Behnken design Pectinase:cellulose:papain- 1:2:1 g/KoH Centrifuged at 11,250× *g*; t: 10 min	Yield: 4.25% DPPH: 1.93%; FRAP: 29.17%; Total phenol: 2543 µg GAE/g; Moisture content: 11.08 g/100 g; Particle size: 8.33 µm pH: 5.23	[149]

EAI—Emulsion activity index, ESI—Emulsion stability index, GAE—gallic acid equivalents, ABTS-2,20-azino-bis(3-ethylbenzothiazoline-6-sulfonicacid) diammonium salt; DPPH—2,2-diphenyl-1-picrylhydrazyl; FRAP—ferric reducing antioxidant power, P—power, F—frequency, t—time, T—temperature, TPC—Total phenolic content, TFC—Total flavanoid content.

**Table 2 foods-11-02973-t002:** Summary of processing protocols applied for encapsulating food materials with ultrasound.

Food Material Encapsulated	Wall Material	O:EM	F (kHz)	P (W)	St (min)	DS (µm)	Effect of Ultrasound	Reference
Encapsulation of Oils and Extracts								
Anatto seed oil	Whey protein isolate Modified snowflake starch	1:4	19	480	5	0.7–1.5 µm	Formation of fine kinetically stable annatto seed oil emulsions. Intensification of the ultrasonication process had no positive effects in the reduction in the droplet size of the dispersed phase.	[177]
Bixin	Cassava starch Corn starch	-	50	150 W	20	-	Ultrasound successfully encapsulated bixin with good encapsulation efficiency. Sonication causes degradation and chemical modification of starch leading to the reducing of particle size, molar mass, and formation of shorter chain.	[178]
Tuna oil	Chitosin Maltodextrin Whey protein isolate	1:2	40	130 and 120	30	0.8–14.1 µm	Sonication improved the stability of tuna oil.	[161]
Soybean oil Mustard oil Pomegranate peel extract	Maltodextrin Whey protein isolate (1:1)	1:30	20	400	5	0–1 µm	Increased sonication time enhances the emulsion kinetic stability, decreasing the droplet size which then improves the emulsifier layering on smaller droplets of W/O emulsion.	[112]
Anthagonin rich extract/grape seed skin/oil	Alginate Pectin beads	20:80	40	200	7	15.3–17.1 µm	Significant decrease in droplet size of beads, alginate beads showed regular, round, and spherical shapes.	[179]
Ginger essential oil/powders	Gum Arabic (GA), Maltodextrin (MD), Inulin (IN)	1:4	20	160	2	GA:MD-6.3 µm GA:IN-5.42 µm GA-6.18 µm	More stable emulsion with smaller droplets. Higher shear stress during sonication leads to a higher level of particles disruption.	[180]
Annatto seed oil	Gum arabic	-	19	800	2	<1 µm	Ultrasonication power reduced the superficial mean diameter and polydispersity and creaming ability of freeze-dried annatto seed oil emulsions.	[65]
Pandan extract	Lysozyme chitosin	1:40 and 1:160	20	150	0.5	1.5 to 2.7 µm	Mean size of the microspheres was generally smallest when using 1 cm probe tip with lower core-to-shell volume ratio but largest when using the 3 mm tip with higher core-to-shell volume, pandan-encapsulated microspheres showed great stability.	[158]
Flaxseed oil	Whey protein concentrate Maltodextrin	1:10	24	150	15	464 nm	Flaxseed oil nanoemulsions helped to enrich muscles with ω-3 fatty acids in broilers by improving blood lipid profile and up-regulation of the hepatic expression of the desaturases and elongates genes. Improved the lipid oxidative stability.	[181]
Starch nano particle (SNP)+peppermint oil loaded SNP	Waxy maize starch	-	20–25	990	5–10	150–200 nm	SNPs showed good uniformity and an almost perfect spherical shape, with diameters of 150–200 nm. The PO-loaded SNPs also exhibited regular shapes.	[182]
Jack fruit extract	Maltodextrin Sucrose ester	-	47	-	5	0.07–7 µm	Submicronic emulsions loaded with jackfruit showed stability for 30 days at 25 °C. Maltodextrin emulsions processed by ultrasound presented a bimodal size distribution with a peak maximum at 138 nm.	[167]
Curcumin in olive oil	Whey protein isolate Primary emulsion Secondary emulsion	1:1	20	750	10	359 nm; 308–386 nm	An increase in the encapsulation efficiency due to enhanced stability of emulsion at higher sonication time. Uniform dispersion of the protein molecules to facilitate their adsorption at the oil-water interface which subsequently increases the retention period of curcumin present in the emulsified droplets.	[183]
Orange peel extract	Alginate chitosin	-	40	150	15	252 µm	Ultrasonication helped in improving the total antioxidant content of orange peel.	[118]
Orange peel using olive oil	Calcium alginate beads	-	40	150	15	0.78 nm	Reduced energy and solvent consumption and ensured high-quality oil. Alginate-oil emulsion was stable and resulted in an EE of 89.5% and oil-loaded beads were spherical with an average size of 0.78 mm. Olive oil successfully applied for ultrasonication of carotenoids from orange peel.	[184]
Butcher Broom Extract	Maltodextrin Gum Arabic	4:1	-	100	15	95.17 µm	Sonication of the extract showed higher DPPH, FRAP, phenolic, anthocyanin, and flavonoid activity.	[185]
Vitamin D	Fish oil	-	20	400	10	300–450 nm	Vit D nanoemulsion stable for a longer period of time as no particle aggregation was observed in stable samples even when left undisturbed for more than 45 days. Improved the EE from 95.7 to 98.2%.	[186]
Essential oil (cumin, basil, clove, calmus, turbose) isoeugenol	Methyl β cyclodextrin	4:1	-	60	30	-	Extending the aqueous solubility and biological properties of the emulsion formed by various essential oils.	[187]
Soybean oil and vit D	Rabbit Bovine liver Oyster Sweet corn	-	20	160	0.75	0.3–8 µm	Sonication of native glycogens formed microcapsules with diameter between 0.3 µm and 8 µm. Sonication helped in improving the protein and glycogen content of soybean oil and vit D emulsion.	[175]
Sacha Inchi oil	Alginate chitosin	-	50	100	-	-	Sonication of Sacha inchi oil showed 30–40% more encapsulation with stable emulsion.	[168]
**Encapsulation of flavor and pigments**								
Cinnamaldehyde and vegetable oil	Chitosin Pectin	7:3				-	Ultrasonicated chitosin and pectin microcapsules exhibited significantly higher cinnamaldehyde retention at high temperatures (>150 °C) and higher flavor retention. Ultrasonication induced amide cross-linking yields more stable polymer bonding.	[169]
*Aspergillus oryzae*	Polyamindo amine dendrine generation 0	-		1.5	1	4 µm	Sonication enhanced ACE inhibitor peptides.	[188]
Peppermint flavor	Gum arabic	-	22	200	-	45.2–255.7 nm	Fine emulsion which enhances the encapsulation and the efficiency of flavor. Ultrasound during encapsulation of peppermint in Gum Arabic, enhances the encapsulation efficiency to 87%.	[189]
Lycopene	Inulin maltodextrin	1:10	20	-	-	-	Sonication of emulsion of lycopene showed higer EE for maltodextrin (78.3%) than inulin. Maltodextrin has high content of encapsulating through lycopene microparticles.	[174]
**Encapsulation of Nutrients**								
Astaxanthin	Alginate Chitosin beads	3:1	40	130 and 220	60 min	4.23 µm	Ultrasonication lead to the appearance of alginate beads with uniform rounded outer surface.	[170]
Oil soluble vitamin (A, D, E)	Raw egg white protein	4:1	20	400	1	5.2 µm	Sonication of egg white proteins stabilized the vitamin D droplets by adsorbing at the oil–water interface. Soluble proteins and small aggregates adsorbed at the oil (vitamin D droplets)-water interface and formed a composite shell likely stabilized by hydrophobic interactions and intermolecular disulphide linkages.	[172]
Vit A	Egg white protein Soy protein isolate CPI	19:1	20	160	1	10–50 µm	Smaller emulsion droplets, better emulsifying properties of proteins that stabilize the oil droplets. SDS-PAGE-sonicated EWP showed a much higher intense band, increase in particle size noted due to denaturation of protein by ultrasonication.	[172]
Soy protein hydrosylate	Nanoliposomes	-	20	100	0.33	193–286 nm	Small unilamellar vesicles by reducing the size of peptide-loaded liposomes 9.5 times in average and increased EE from 2 to 13% to 4–20%, improved the stability of nanoliposomes. Sonication showed a positive correlation with the *EE* of soy protein, indicating that no thermal degradation of phospholipids occurred.	[190]
Prebiotic Lactic acid bacteria, bioactive γ-aminobutyric acid	Dextran Whey protein	20:80	20	80	0.5–2.0	LAB-3 to 5 µm GABA-1.5 to 3 µm	Ultrasound parameters of LAB and GABA were refined to optimize the size of the W/O droplets and outer microcapsules (5–15 μm). Ultrasound treatment for up to 200 s had no significant effect on bacteria viability (>109 CFU/mL).	[171]
*Lactiplantibacillus plantarum* BCRC 10357	Calcium alginate	-	40	300	15	-	β-glucosidase activity of encapsulated *L. plantarum* was 11.47 U/mL at 12 h, then increased to 27.43 U/mL at 36 h and to 26.25 U/mL at 48 h in black soymilk at 37 °C, showing the high adaptation of encapsulated *L. plantarum* encountering ultrasonic stress to release high β -glucosidase.	[191]
Omega-3 fatty acid	DPA 10% EPA 42%	-	25	-	30–60	-	Ultrasound application helped in increasing the amount of healthy fatty acid in pork meat.	[192]

(F—frequency, P—power, EE—encapsulation efficiency, O:EM—oil: encapsulating material, DPPH—2,2-diphenyl-1-picryl-hydrazyl-hydrate, FRAP—ferric reducing antioxidant power, ACH inhibitors- acetylcholinesterase inhibitors, LAB-Lactic acid bacteria, GABA-γ-aminobutyric acid, LAB-Lactic acid bacteria, SDS PAGE—Sodium dodecyl-sulfate polyacrylamide gel electrophoresis, EWP—egg white proteins, DPA—Docosahexanoic acid, EPA—Eico sapentanoic acid).

## Data Availability

Not applicable.

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
