# Peer review of "Ultrasound-Assisted Extraction and the Encapsulation of Bioactive Components for Food Applications"

_foods, 2022, doi:10.3390/foods11192973_

Round 1
Reviewer 1 Report
foods-1891077-peer-review-v1
The authors presented a review paper on “Ultrasound assisted extraction and encapsulation of bioactive components for food applications” with the aim to discuss ultrasound technology, its applications, fundamental principles of ultrasonic assisted extraction and encapsulation, parameters affecting them and applications of ultrasound assisted extraction and encapsulation in food systems. The review article provides the knowledge and emphasis for ultrasonication as extraction and encapsulation technology. Recently, ultrasound assisted extraction (UAE) is gaining popularity due to relative advantages over solvent extraction. It is a non-thermal or green technology which can lower cost of operation, decrease operational time and energy consumption as compared to conventional methods.
Based on this, I have a minor comment:
1. In general, the novelty of the review is missing.
2. Line 133 says “Ultrasound assisted extraction primarily involves interaction of ultrasonic power and solvents to extract targeted bioactive components from the plant biomass”. If it use solvent in addition to ultrasonic power, then why you stated it as a green technology and it has an advantage than the conventional extraction methods (for example solvent extraction).
3. In Figure 3, please add its disadvantage or drawback (you can refer this paper: DOI:10.3390/ijerph18179153).
4. In section 4.2, it is important to discuss about physical parameters that can affect the power of the ulstasonication and if possible support it with the equation and even review some literatures about the physical parameter effects on the yield and quality of the extracts in a tabular form.
5. In section 4.4, the authors indicated that ethanol and acetone can be used for extraction phenolic compounds. Is it safe to use acetone as a solvent for extraction?
6. Section 6.1 is about the UAX of proteins and lipids. The authors discussed about the extraction of proteins. The UAE of lipids is missing. So, please incorporate it.
7. There are some typos. Please correct them.
Reviewer 2 Report
The article „Ultrasound assisted extraction and encapsulation of bioactive components for food applications“ is a review article focused on an overview of the promising applications of ultrasonic techniques in functional food preparation.
Ultrasonic techniques are very popular and have a big potential in food industry because they meet new trends. So, I it is a good idea to review this topic. But there are some issues in this paper which must be fixed before the potential publication. For first, I strongly recommend an English revision of your current manuscript. Additionaly, some sections need fixing.
Generally, many sections are not described properly. It seems to me that there are some descriptions in the beginning and then are the same/similar information in extensive tables.
The division of chapters also does not seem very well thought out – e.g. 8.1. Encapsulation of oils and extracts ...then 8.3. Encapsulation of nutrients ....oils are not nutrients?
For better clarity, in 4.1 section I suggest to add a table with presented studies – e.g: frequency, purpose – extraction/encapsulation, ... and then to evaluate which frequencies are the best for extraction/encapsulation...
line 250 – „certain level“ – has to be specified
section 260 – 275 – add reference
line 287 – which studies – references? values?
line 322 – which temperature? specify a value +323 - 325 – how were the temperature here?
section 4.4. – I suggest to add some information – for example to discuss water extraction, which I think is the most popular now
line 368 – values again
line 396 – reference
line 632 – many researchers – references
Round 2
Reviewer 1 Report
Dear authors,
Thank you for correcting all the comments I raised. Now, the review manuscript is fine for me and can be published as it is.
Reviewer 2 Report
Authors have improved their manuscript. I agree with the current version.